# Habitat use by female desert tortoises suggests tradeoffs between resource use and risk avoidance

Melia G. Nafus[1¤], Jacob A. Daly[2], Tracey D. Tuberville[2], A. Peter Klimely[1], Kurt A. Buhlmann[2], Brian D. Todd[1]*

**1** Department of Wildlife, Fish and Conservation Biology, University of California, Davis, Davis, CA, United States of America, **2** University of Georgia's Savannah River Ecology Lab, Aiken, SC, United States of America

¤ Current address: US Geological Survey, Fort Collins Science Center, Fort Collins, CO, United States of America

* btodd@ucdavis.edu

**Data Availability Statement:** All relevant data can be found here: Nafus MG. (2022) Activity and habitat selection by female desert tortoises in Mojave National Preserve, 2011 – 2013: U.S.

## Abstract

Animals may select habitat to maximize the benefits of foraging on growth and reproduction, while balancing competing factors like the risk of predation or mortality from other sources. Variation in the distribution of food resources may lead animals to forage at times or in places that carry greater predation risk, with individuals in poor quality habitats expected to take greater risks while foraging. We studied Mojave desert tortoises (*Gopherus agassizii*) in habitats with variable forage availability to determine if risk aversion in their selection of habitat relative was related to abundance of forage. As a measure of risk, we examined tortoise surface activity and mortality. We also compared tortoise body size and body condition between habitats with ample forage plants and those with less forage plants. Tortoises from low forage habitats selected areas where more annual plants were nutritious herbaceous flowering plants but did not favor areas of greater perennial shrub cover that could shelter them or their burrows. In contrast, tortoises occupying high forage habitats showed no preference for forage characteristics, but used burrows associated with more abundant and larger perennial shrubs. Tortoises in high forage habitats were larger and active above ground more often but did not have better body condition. Mortality was four times higher for females occupying low forage habitat than those in high forage habitat. Our results are consistent with the idea that tortoises may minimize mortality risk where food resources are high, but may accept some tradeoff of greater mortality risk in order to forage optimally when food resources are limiting.

## Introduction

Forage quality, availability, and species composition can affect habitat use by herbivores [1, 2]. Animals foraging optimally will generally maximize their net energy intake per unit of time [3]. Increased net energy intake through time can yield higher growth rates [4], greater

Geological Survey data release, https://doi.org/10.5066/P9NBNE08.

**Funding:** National Science Foundation Graduate Research Fellowship Program under Grant No. DGE-1148897 (MGN) California Energy Commission's Public Interest Energy Research (PIER) Program under Agreement # 500-10-020 (BDT) USDA National Institute of Food and Agriculture, Hatch project CA-D-WFB-2097-H (BDT) Department of Energy under Award Number DE-FC09-07SR22506 (TDT, KAB) Department of Energy under Award Number DE-EM0005228 (TDT) US National Park Service under PSAC-CESU Cooperative Agreement number P17AC01606 (TDT, BDT, KAB) The funders had no role in study design, data collection and analysis, decision to publish, or preparation of the manuscript.

**Competing interests:** The authors have declared that no competing interests exist.

reproductive output [5, 6], and reduce risk of starvation. Because variation in abiotic conditions across space and time results in habitat patches that can differ in quality, individuals should choose habitat based on their capacity to optimize foraging success. However, exposure to adverse conditions or predation risk can also affect foraging behavior and thus habitat use by imposing costs to survival [7].

How prey behave in response to the presence of predators can be an important force structuring space use and community function [8, 9]. Prey may alter their foraging behavior, and, to a lesser extent, their habitat use in response to predation risk [10–12]. Consequently, individuals can exhibit behaviors that reflect tradeoffs between energy acquisition and risk avoidance [13–15]. Individuals are expected to forage such that the lowest ratio of mortality to gross foraging rate is achieved [16]. Risk avoidance thus depends on context and can also be affected by how urgently an individual needs food [17–19]. Specifically, individuals operating on an energy surplus should avoid risky behavior, whereas individuals experiencing energy deficits must improve their energetic state even when doing so requires riskier behavior [7]. Risk-taking can thus be "state-dependent," whereby individual condition influences behavioral choices and habitat use [20, 21]. Tradeoffs between foraging and risk avoidance are thus likely to be highly spatially and temporally variable as individual body condition fluctuates in response to changing resource availabilities.

Spatial and temporal resource variation are key characteristics of desert ecosystems [22]. Plant productivity depends on seasonal and annual precipitation patterns [23]. Precipitation is both annually variable and patchily distributed across the landscape [22], which can result in considerable variation in forage availability over small spatial scales. Mojave desert tortoises (*Gopherus agassizii*), native to southwestern deserts of the United States, can have large fluctuations in body mass in response to resource conditions [24], and their activity and behavior are tightly linked to resource availability. Tortoises are typically active during the spring and late summer [25–27], when the forbs, or herbaceous plants, and grasses on which they feed are flowering [28]. Although desert tortoises are sometimes characterized as selective foragers, the degree of selectivity can depend on local conditions [28, 29].

Here, we examined habitat use by Mojave desert tortoises in relation to the availability of forage plants or large perennial plants that can serve as shelter. We also compared surface activity, body size, body condition, and mortality between tortoises from habitats with either high or low forage availability. We sought to determine whether observational patterns in habitat selection by desert tortoises at our study site were consistent with predictions framed under risk tradeoff theories. We predicted that tortoises in high quality forage patches would select habitat that prioritized large shelter plants, whereas tortoises in low quality forage patches would select habitat that prioritized access to nutritious forage plants. The anticipation of a tradeoff between these choices led us to predict increased mortality risk, smaller body size, (a measure of long-term resource availability) and less activity in lower quality habitat [30], while maintaining similar body conditions.

## Materials and methods

### Study area

Our study was conducted in Ivanpah Valley within Mojave National Preserve, California, USA (34°53′N, 115°43′W). Ivanpah Valley is located in the eastern portion of the Mojave Desert (east of the 117°W meridian line), which has a bimodal precipitation pattern [31]. Annual forbs and grasses in Ivanpah Valley typically flower in the early spring and late summer in response to precipitation, but inter-annual timing and extent of their blooms are highly variable [23]. Perennial structure of Ivanpah Valley can be generally categorized as

Mojavean-Sonoran Desert Scrub. Within our ~30 km² focal study area, we divided perennial communities into two general habitat types based on vegetation communities described by the Hierarchical List of Natural Communities with Holland Types [32]. "Yucca woodland" (YW) habitat was composed of Mojave yucca scrub (*Yucca schidigera* Alliance) or Joshua tree woodland (*Yucca brevifolia* Alliance) [Fig 1A]. "Creosote scrub" (CS) habitat was generally creosote bush scrub (*Larrea tridentata* Alliance) or Creosote bush–white bursage scrub (*L. tridentata-Ambrosia dumosa* Alliance) [Fig 1B]. The focal study area had an east-west elevation gradient from 800–1050 m. Both perennial communities occurred across all elevations, but YW habitat tended to be more prevalent at elevations >950 m. Both annual and perennial plant resources varied between the two habitats (described further below in Methods and Results sections), with the YW habitat having more abundant annual plants to forage upon than the CS habitat.

## Study animals

Male Mojave desert tortoises range widely to seek mates during foraging seasons. Thus, fine-scale habitat selection by males may partly reflect their maximizing mating opportunities, whereas female habitat use may more accurately reflect choices about foraging opportunities and thus, environmental conditions. For this reason, we used adult female desert tortoises to study habitat selection over a series of assessments that were done over multiple temporal scales from 2011–2013. However, females are typically gravid in the spring and early summer, and thus, habitat selection may be influenced by nest site selection and not solely by resource acquisition. Although female tortoises are likely procuring resources for the next reproductive season in August, they are not nesting. Therefore, we categorized forage in association with burrows during the summer season in which forage was available but nesting sites should not be driving burrow selection.

Between March 2011–June 2012, we affixed 30 females with radio transmitters (20 g, model RI-2B, Holohil Systems Ltd., Carp, Ontario, Canada) on their first left or right costal scute and individually marked them by filing their marginal scutes [33, 34]. We tracked females weekly during their active seasons, monthly when dormant, and rotated time of day when each female was tracked. At each encounter, we recorded location using a global positioning system unit (Garmin eTrex 20 [± 3 m], Schaffhausen, Switzerland). Females were categorized as occupying one of the two habitat groups (YW or CS) based on the dominant perennial community structure in their home range. We categorized 13 females as CS and 17 females as YW. Handling of all tortoises followed protocols outlined in permits provided by US Fish and Wildlife Service (Permit # TE-17838A), California Department of Fish and Wildlife (Permit # SC-11072), and Mojave National Preserve (Permit # MOJA-00258), and procedures approved by the Institutional Animal Care and Use Committee through the University of California, Davis (IACUC # 15997).

## Tortoise morphometrics, activity, and survival

In September 2012, we recaptured females and measured their mid-line carapace length (MCL, distance from the nuchal scute to the pygal scute), carapace width (CW, maximum width at the third vertebral), and carapace height (CH, centered on the third vertebral), to the nearest mm and we recorded mass to the nearest 50 g. We used shell measurements to calculate volume using a modified formula for an ellipsoid ():[35]

$$\text{Volume} = \frac{\pi * \text{MCL} * \text{CW} * \text{CH}}{6000} \text{cm}^3$$

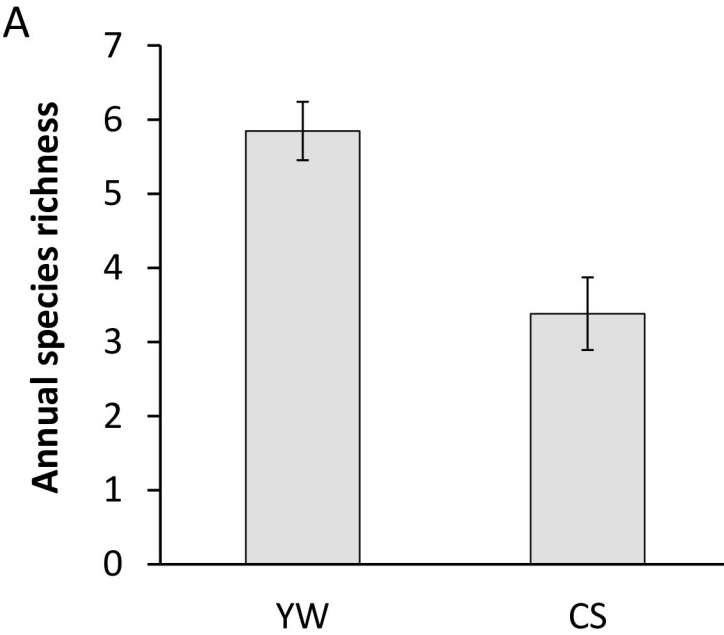

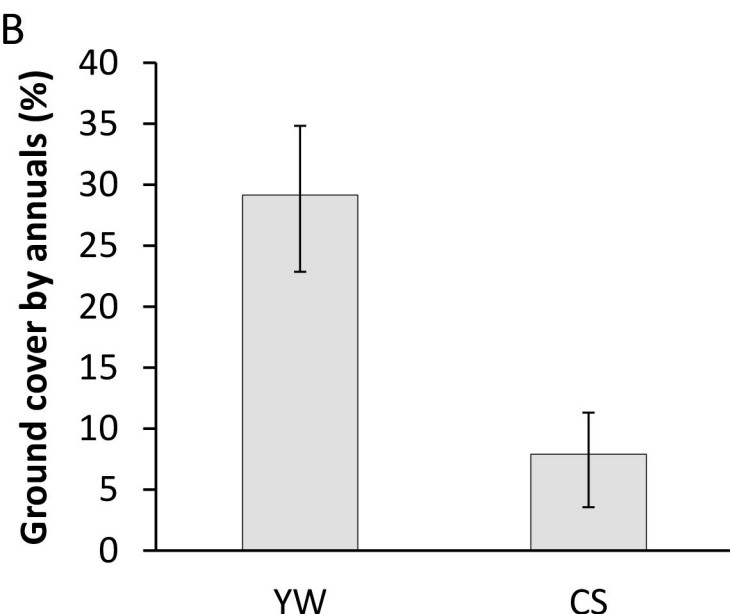

**Fig 1. Images characterizing study habitat.** Images of creosote scrub (CS; A) and yucca woodland (YW; B) habitats in Ivanpah Valley of Mojave National Preserve, California, USA. Both pictures were taken on the same day in August 2012.

from which we calculated female body condition ():[36]

$$Body\ Condition\ Index = \frac{Mass}{V_{Shell}}\ g/cm^3$$

Due to unequal variances, we used Welch's nonparametric two-sample *t*-test in R 3.1.1 [37] to test for differences in MCL and body condition between CS and YW females. We accepted significance at $\alpha < 0.05$ and report morphometric values as means (± 1 SE). We measured surface activity of radio-tracked tortoises by recording whether a female was 'not visible' (deep inside a burrow) or 'visible' (either at the entrance of her burrow or on the surface) at each tracking event from March 2011 –December 2013. We used a two-sample test of proportions to compare proportion of not visible encounters between YW and CS females. To test for differences in survival between the habitats while accounting for differences in monitoring periods for each animal we used a Cox Proportional Hazard regression model in the survival package in R 3.1.1 [37] for the same time period. We included habitat site and season as predictors in the model, where season was defined as the active season (spring to late summer) and dormant season (fall through winter).

## Tortoise habitat selection and space use patterns

To compare tortoise habitat use versus availability in each area, we collected annual plant data near females' locations during 15–30 August 2012 (CS, *n* = 11; YW, *n* = 17). We collected data on both annual and perennial plants, both of which are important resources for desert tortoises—annual plants as forage and perennial plants as refuge [38]. We measured annual plant communities using quadrats and we used line-point transects to measure perennials near occupied burrows. We used occupied burrow locations as the origin points for these habitat data collections because burrows are sites where tortoises spend much of their time. In contrast, surface locations of tortoises may only represent movement across the landscape. We generated a random control location (Random Point Generator, GeoMidpoint) for each tortoise burrow location, placed exactly 200 m away from the occupied burrow (control; CS, *n* = 11; YW, *n* = 17), to characterize available habitat.

We generated ten random points within 50 m of each burrow and control location (Random Point Generator, GeoMidpoint). We selected 50 m due to observations that females at our sites often foraged within 50 m of their burrow. At each point, we placed a 1 m² quadrat (Fig 2) within which we recorded annual species richness (number of species), stem counts (number of independent stems) of forbs or grasses, and ground cover (%). We counted multiple stems as a single plant if the stems bifurcated after exiting the ground. Because annual forbs may have more nutritional value for tortoises than grasses [39, 40], we also calculated the proportion of annual stems that were grass (i.e., the fraction, by stem count, of the total number of annuals that were one or more annual grass species) or forbs.

In an effort to ensure that we collected annual plant data for all females during the brief period in which they were flowering, we measured perennial vegetative cover at the occupied August burrow and control locations several weeks later (04–07 November 2012). Because woody perennial plants do not change during such a short period, the difference in sampling time was not meaningful with regard to tortoise ecology. We used six, 10-m line-point transects that radiated from the occupied burrow or paired point (Fig 2). We documented the total number and species of woody perennials that touched the transect line to estimate perennial abundance and species richness. Perennial shrubs were considered individual plants if there was 0.5 m between two plant bases. In addition, for the three perennials nearest to the control or burrow location, we measured the length (cm), width (cm), and height (cm), from which we calculated their volume using a modified formula for a sphere:

$$VP = \frac{\pi * (\frac{l*w*h}{3})^3}{6} \, cm^3$$

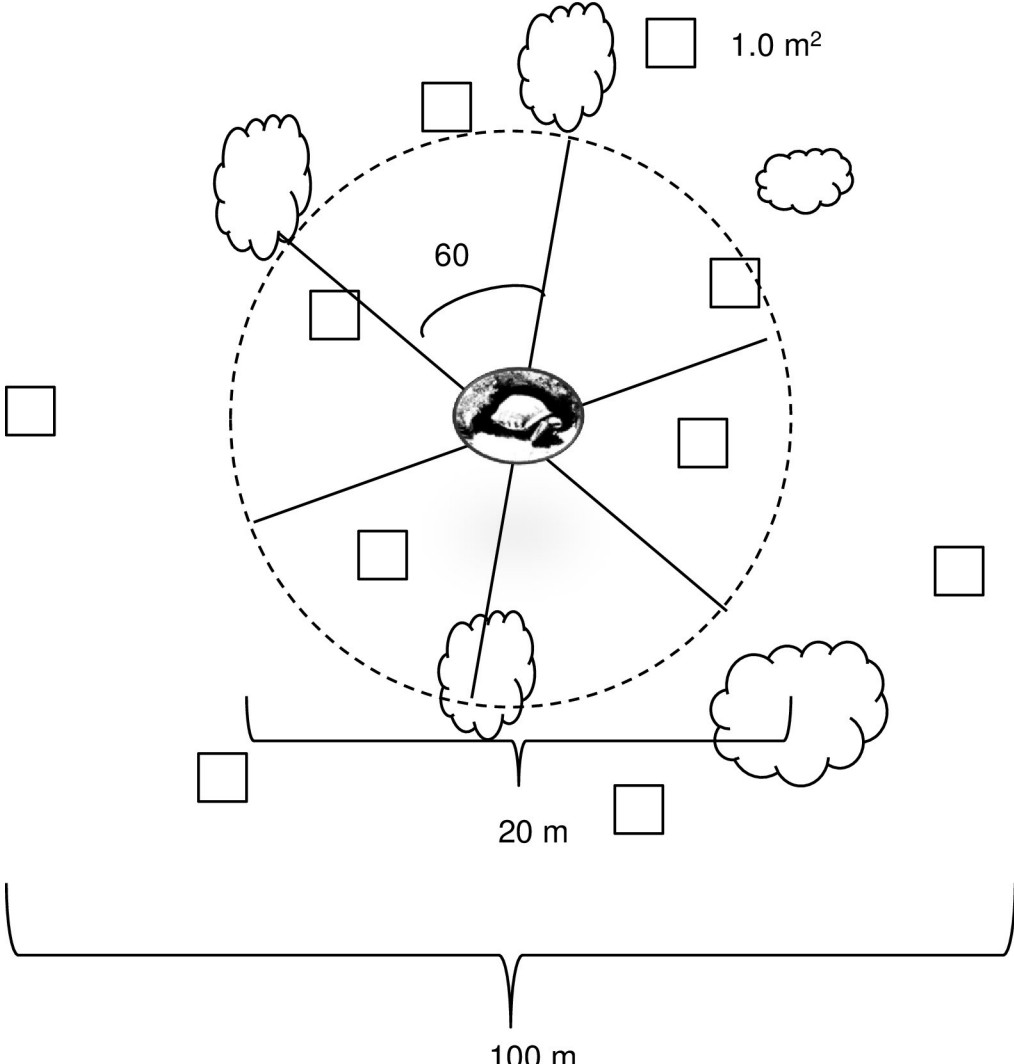

**Fig 2. Habitat sampling design.** The habitat sampling design for used (female tortoise burrow) or available (paired control) habitat sampling locations in Mojave National Preserve. We created a transect array composed of six (10 m) transects to measure perennial cover. Within a radius of 50 m, we sampled annual vegetation at 10 randomly selected points using a 1 m² quadrat.

For each sampling location, we took the mean value for each vegetation variable to create a single metric to use in analyses. We used two independent logistic regression models with a binomial distribution to compare used (score = 1) and available (score = 0) habitat by females in CS and YW. Prior to model development, we tested for multicollinearity among the habitat variables using variance inflation factors, for which none were observed. We included woody perennial volume, perennial abundance, perennial species richness, annual species richness, percent ground cover by annuals, and percentage of annual stems that were forbs as predictor variables. As we used a paired sampling regime, we also included female identification number as a random effect.

Due to the short season of the summer blooms and other field commitments, we only collected a single habitat sample for each female in August. Thus, instead of completing repeated

measures for each female, we calculated the proportion of encounters that occurred within 50 m of the August-occupied burrow during the seven-week foraging season (15 July– 01 September 2012). Our goal was to determine the relative value of the defined area (within 50 m of the burrow) as a foraging patch during summer 2012.

## Characteristics of annuals in the two habitats

We used annual plant data from our control plots to compare forage availability and/or quality between YW and CS habitats in Ivanpah Valley. We used one-way ANOVA in R [37] to compare stem numbers and species richness. To compare percent ground cover and proportion of annual stems that were grass, we used general linear models with Poisson and binomial distributions, respectively. Because tortoises can be selective foragers, we also opportunistically documented which annual species tortoises were seen eating, hereafter referred to as "consumed" species. We did not complete bite counts or other in-depth foraging studies. Thus, consumed plant species were merely those that we documented tortoises at our site eating and may not represent the entirety of their diet or a measure of preference. We used Pearson's product-moment correlation coefficient in program R to measure the correlation between the stem counts of consumed species and annual species richness, total annual stem counts, or percent ground cover by annuals. All data are available to download [41].

## Results

### Tortoise morphometries, activity, and survival

We found small females in both CS and YW habitats, but the largest females were absent from CS, the low forage habitat. The average MCL of all females encountered in YW, the high forage habitat, was 236 ± 3 mm and ranged from 217–265 mm. Females located in CS habitat, in contrast, had an average MCL of 229 ± 3 mm and ranged from 211–247 mm, which was significantly smaller ($t$[29] = -1.74, $P$ = 0.04). There was no significant difference in body condition between the two habitats at the close of the foraging season in September 2012 ($t$[27] = 0.33, $P$ = 0.37). Females in the two sites also did not differ in surface activity ($\chi^2$ = 27, $P$ = 0.41) during the period of our study.

From 2011–2013, four females died and one female was lost as a result of probable radio failure, yielding overall survival of 86% for the 29 known-fate females over three years. Mortalities were evenly distributed across years (Table 1). Overall mortality was nearly four times higher in CS (3 of 11 females; 23%), the low forage habitat, than in YW (1 of 17 females; 6%; $\beta$ = 1.4, SE = 0.5, $z$ = 2.5, $P$ = 0.02), the high forage habitat. Season was not significantly correlated with mortality ($\beta$ = 5.7, SE = 3.0, $z$ = 0.002, $P$ > 0.10), but overall mortality was low.

**Table 1. Annual survival for female desert tortoises.** Female tortoises were monitored using radio-telemetry over a three-year period in creosote scrub (CS) and yucca woodland (YW) habitats in Mojave National Preserve, California, USA.

| Annual survival | | | | | | |
|---|---|---|---|---|---|---|
| Year | CS | | | YW | | |
| | *n* | *deaths* | Survival | *n* | *deaths* | Survival |
| 2011 | 10 | 1 | 0.90 | 8 | 0 | 1.00 |
| 2012 | 11 | 1 | 0.91 | 17 | 1 | 0.94 |
| 2013 | 11 | 1 | 0.90 | 16 | 0 | 1.00 |
| **MEAN** | | | **0.90** | | | **0.98** |

## Tortoise habitat and spatial use patterns

Females in CS selected different habitat characteristics than did females in YW. CS females occupied burrows that were adjacent to slightly greater percentages of forb stems (91 ± 2%) than were available at random points (81 ± 5%). Percentage of forbs was the only parameter significantly correlated with August-occupied burrows in CS, while no other habitat characteristic correlated with August-occupied burrows (Table 2). In contrast, YW female August-occupied burrows were characterized by larger perennials (620,435 ± 141,151 cm$^3$ versus 335,427 ± 76,716 cm$^3$) at greater densities (6 ± 0.4 versus 4 ± 0.3 perennials) than were randomly available (Table 2). We found no difference in the community of annual plants within 50 m of YW burrows compared to controls (i.e., in used vs. paired habitat) (Table 2). During the seven-week summer foraging period (15 July– 01 September 2012), females in both habitats were located within 50 m of the August occupied burrow 53 ± 4% (range: 20–100% of the observations).

## Characteristics of annuals in the two habitats

Within the control quadrats, all annual species present in YW were also present in CS, but structure and heterogeneity differed between the two habitats. Mean species richness per plot was significantly greater at 5.8 species (95% CI, 5.5–6.2) in YW versus 3.4 species (2.9–3.9) in CS ($F_{1,26}$ = 65.1, $P$ <0.001; Fig 3A). Ground cover by annuals was 29% (23–35%) in YW and 8% (5–12%) in CS ($\beta$ = 1.2, SE = 0.1, $z$ = 11.2, $P$ < 0.001; Fig 3B). Annual grasses, in the form of *Bouteloua astridoides* and *B. barbata*, comprised 56% (47–65%) of the annual stems per plot in YW, and accounted for 19% (8–29%) of stems in CS plots, which approached statistical significance (*β = 1.7, SE = 0.9, z = 1.8, P < 0.06*, see species list in S1 Table). Overall total availability of forbs was nearly double in YW than in CS, with 54 (38–71) stems compared with 27 (15–44) stems ($F_{1,26}$ = 5.7, $P$ = 0.024). The mean abundance of consumed species in YW (68 [45–104] stems), was 34 times greater than in CS (2 [1–3] stems; $F_{1,26}$ = 122, $P$ < 0.001). Stem density of consumed species was positively correlated with annual plant species richness ($r$ = 0.46, $n$ = 56, $P$ < 0.001), ground cover ($r$ = 0.75, $n$ = 56, $P$ < 0.001), and total stems ($r$ = 0.95, $n$ = 56, $P$ < 0.001; Fig 4). Thus, species richness, ground cover, and stems of annual vegetation reflected abundances of plant species known to be consumed at this study site (see S1 Table for a list of annual plants detected).

**Table 2. Vegetation characteristics from female tortoise habitat around burrows occupied by females or paired points in August 2012 at Mojave National Preserve.** Results of linear regression of habitat characteristics on selected versus available habitat in yucca woodland (YW) and creosote scrub (CS). Means are of habitat variables measured near occupied burrows. Significance indicated by '*'.

| Variable | YW | | | CS | | |
|---|---|---|---|---|---|---|
| | Mean ± SE | β | Pr(>\|t\|) | Mean (± SE) | β | Pr(>\|t\|) |
| Perennial | | | | | | |
| Volume (cm$^3$) | 621,674 ± 141,151 | 4.1e-7 | 0.01* | 510,324 ± 103,849 | 2.8e-7 | 0.48 |
| Abundance | 6 ± 0.4 | 0.2 | 0.001* | 4 ± 0.5 | -0.2 | 0.10 |
| Species Richness | 5 ± 0.3 | 0.1 | 0.69 | 2 ± 0.1 | 0.7 | 0.07 |
| Annual | | | | | | |
| Ground Cover | 32 ± 3 | 6.2e-3 | 0.43 | 13 ± 2 | -9.5e-3 | 0.64 |
| Stems | 143 ± 21 | -2.7e-4 | 0.83 | 40 ± 6 | -8.6e-3 | 0.16 |
| Species Richness | 6 ± 0.1 | 0.2 | 0.11 | 4 ± 0.3 | 0.2 | 0.37 |
| Percent Forb | 63 ± 4 | -0.6 | 0.34 | 91 ± 2 | 2.4 | 0.01* |

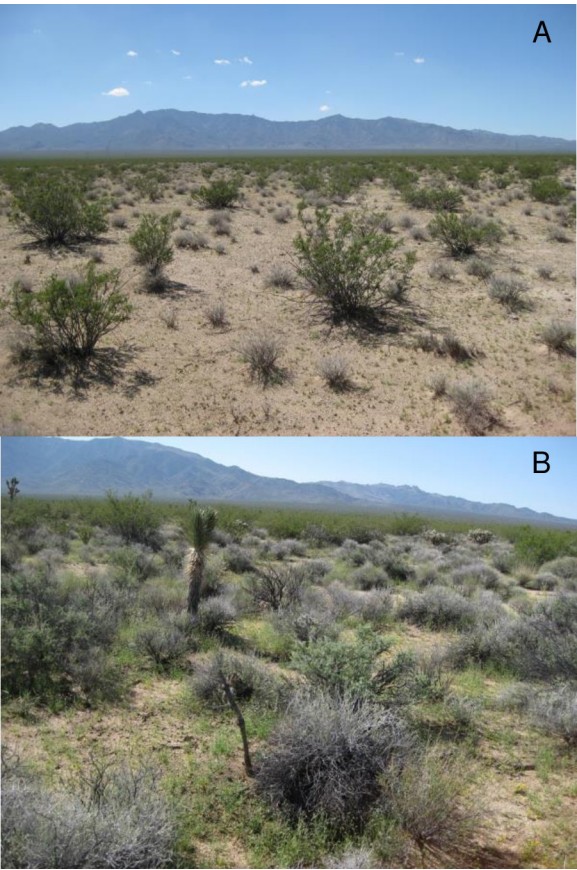

**Fig 3. The species richness and ground cover by annual plants in Mojave National Preserve in August 2012.**
Annual species richness (A) and percent ground cover (B) in summer 2012 by site presented as means with 95%
confidence intervals. Yucca woodland (YW) had greater species richness and ground cover by annuals per plot than
did creosote scrub (CS; *P* < 0.001 in both cases).

## Discussion

Female tortoises in this observational study used habitat in a manner consistent with context-
dependent risk appraisal given differences in local forage quality. Specifically, tortoises in habi-
tat with less abundant forage used burrows closer to nutritious forage plants than expected at
random, whereas tortoises in habitat with abundant forage used burrows more sheltered by
large perennials than expected at random. Females in both locations had equivalent body con-
dition scores during the period of study, but not differences in surface activity. The summer
foraging season habitat use aligned with subsequent predictions about body size and mortality
over the three year period of study, if habitat use patterns were temporally consistent. Repeat-
ing this study using populations from other parts of the species' range would be useful to
understand how broadly these results may apply to risk appraisal and habitat use in desert tor-
toises. Despite being limited to one population within one season of study, the findings here
do suggest that risk-taking behavior in wildlife can depend on resource conditions experienced
by individuals. By varying the risk associated with habitat selection, female tortoises in low for-
age habitat were able to maintain similar body conditions as individuals in high forage habitat
during the period of study.

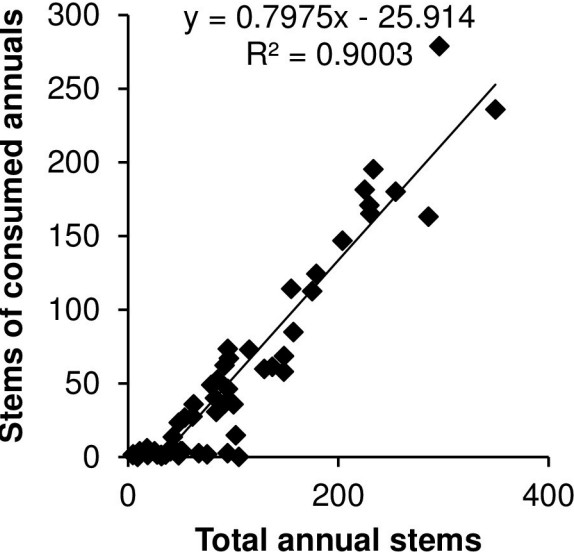

**Fig 4. Correlations between total annual abundance and abundance of consumed annual plants in Mojave National Preserve during August 2012.** We categorized forage into consumed and non-consumed forage through observations of telemetered tortoises. We found a strong positive relationship between the total annual abundance and the abundance of one or more consumed species ($r = 0.46$, n = 56, $P < 0.001$).

### Implications of tortoise size differences between habitats

We encountered small female desert tortoises in both habitats, but large females were absent from CS habitat. The absence of large tortoises in an area can be interpreted as evidence they are not surviving long enough to reach large sizes concomitant with older age [42], an interpretation consistent with tortoises prioritizing access to limited forage plants instead of large perennial plants that can shelter them from harm. In general, desert tortoises are thought to have high inter-annual site fidelity [43–46]. Recent studies in turtles also indicate that size differences can arise from variation in age at maturity and somatic growth rates in early life [47, 48], with large adults having experienced better resource conditions when young than did small adults [49, 50]. Although we have no information on the historical home ranges of the individuals in this study, it could be possible that female tortoises in low forage CS habitat may be moving to other habitats as they grow and age to avoid predation or otherwise improve survival.

Differing rates of somatic growth among Testudines can be caused by differences in food availability. In many taxa, adult body size is closely tied to historical resource availability, with larger animals originating from habitats with greater food availability [51]. In reptiles, somatic growth rates positively correlate with food availability [4]. Desert tortoises, in particular, have greater rates of growth when provided with better quality or greater quantities of food [52, 53]. Historically consistent differences in resource availability between the two habitats could thus yield different somatic growth rates or adult body sizes in the females living there [49], although 2012 was considered a drought year at this study site. Differences in forage availability and thus behavioral effects may have been more pronounced during a drought [54]. Within that period, however, we found evidence that the YW habitat provided greater forage potential for desert tortoises. Furthermore, as adult desert tortoises often experience slow to negligible rates of somatic growth [55], we were unable to determine whether greater abundances or quantities of forage or even equivalent body condition in the short term would translate into increased rates of growth and thus large body size in the long term.

An alternative explanation for the observed differences in body size between the habitats could be that females were in greater abundance in one habitat, and territoriality or intra-specific competition may have resulted in slower growth in the low forage habitat. Larger females might exclude smaller females from high quality YW habitat, if territoriality limits co-occupancy. Female-female aggression and burrow defense have been reported in the closely related gopher tortoise (*Gopherus polyphemus*) [56]. Space use by gopher tortoises—including home range size, overlap, and burrow sharing—also varies as a function of population density [57]. It is unknown whether social behavior is as important in *G. agassizii* populations, which tend to occur at lower densities than gopher tortoises, but it is possible that aggressive behavior may have occurred between females of different sizes. However, in multiple cases during this study the authors also observed females sharing burrows without aggression. A future comparison of growth rates in juvenile tortoises occupying each habitat and the relative availability of food could provide greater insight into a potential mechanism behind the size differences we observed.

## Trade-offs between risk and resource acquisition

We found evidence that micro-scale habitat use differed between two habitats with differing foraging opportunity for tortoises. Resident females in the high forage YW habitat showed no evidence for selection of forage conditions—perhaps because forage was not limiting—but did choose habitat with large shelter perennial plants. In contrast, females occupying CS habitat preferred areas with more nutritious forbs than grasses, but did so at the cost of being unable to stay close to large shelter perennial plants. Selective use of habitat with enhanced forb availability may have helped CS females maintain comparable body conditions to those from the YW habitat that had overall greater foraging opportunities. Altering foraging strategy based on local conditions can allow individuals in habitats with poor forage conditions to forage with comparable efficiency to those in habitats with more abundant resources [58]. Ultimately, patchy or limited resources can encourage selective use of areas that increase access to forage [59, 60]. Thus, preferential use of areas with greater concentrations of forb species may yield increased foraging efficiency. Given their limited activity periods (~ 3 hrs per day [26]), desert tortoises located in forage-limited habitats could optimize foraging behavior by minimizing time spent finding food and maximizing energy obtained when feeding. To so, females may have accepted greater predation or thermal exposure risk. However, with the resolution of data we were able to collect, we saw no difference in surface activity by females. In prior work on juveniles, individuals with greater access to water and food spent more time on the surface [30]. Thus surface activity may not be a direct measure of risk if not paired with other measures, such as ground temperature or immediate presence of a predator. Risk-avoidance can be context-dependent and partially dependent upon individual state [17–19], such that individuals with an energy surplus avoid risk relative to those with energy deficits [7]. Consistent with this explanation is that females in the lower forage area had higher risks of mortality over the three years of monitoring we conducted. Mortality may have been caused by predation or exposure, both of which would be anticipated to increase with tortoises making use of more unsheltered habitat.

## Conclusions

The observed use of habitat by female tortoises in this study was consistent with expected tradeoffs between resource acquisition and risk avoidance given the differences in forage quality between the two habitats. Our interpretation of the data is that selecting for annual forage—as opposed to prioritizing greater perennial cover—was partly successful for females in habitat

with lower forage availability because they were able to maintain similar body conditions to females that had greater access to forage. However, the female tortoises in the lower quality forage habitat were of smaller body sizes and experienced higher mortality than females from the higher forage quality habitat. Although consistent with such a tradeoff, our findings are based on a relatively small number of individuals at a single site over a limited period and are thus limited in scope. In addition, mortality was low during the study period. Thus, further study is needed to support any assertion that habitat quality could influence habitat selection by desert tortoises at fine spatial scales and, in turn, affect morphometric parameters and mortality risk. We argue that female behavior supported that behavioral plasticity can enhance ability to withstand short periods of drought or other resource limitations by prioritizing different habitat characteristics (food or shelter). This study, incipient in nature, should ideally lead others to examine the cost-benefit balance between refuge seeking and foraging activities in a species whose habitat can vary considerably across space and time.

## Supporting information

**S1 Table. Ephemeral forb and grass species documented in 1 m 1 m habitat plots during summer 2012 (August 15–30) in Mojave National Preserve.** This is a complete list of all species that were recorded to be present during the sampling of habitat around Mojave desert tortoise occupied burrows and a paired point within 200 m of the burrow.
(DOCX)

## Acknowledgments

We especially thank T Esque, JP Rose, and several anonymous reviewers for providing feedback on earlier versions of this manuscript. Additionally, we thank JM Peaden for assistance tracking female tortoises. We also thank National Park Service staff at Mojave National Preserve for their intellectual support. Any use of trade, firm, or product names is for descriptive purposes only and does not imply endorsement by the U.S. Government.

## Author Contributions

**Conceptualization:** Melia G. Nafus.

**Data curation:** Melia G. Nafus.

**Formal analysis:** Melia G. Nafus.

**Funding acquisition:** Melia G. Nafus, Tracey D. Tuberville, Kurt A. Buhlmann, Brian D. Todd.

**Investigation:** Melia G. Nafus.

**Methodology:** Melia G. Nafus, Brian D. Todd.

**Project administration:** Melia G. Nafus, Tracey D. Tuberville, Brian D. Todd.

**Resources:** Melia G. Nafus, Brian D. Todd.

**Supervision:** Tracey D. Tuberville, A. Peter Klimely, Brian D. Todd.

**Validation:** Melia G. Nafus.

**Writing – original draft:** Melia G. Nafus.

**Writing – review & editing:** Melia G. Nafus, Jacob A. Daly, Tracey D. Tuberville, A. Peter Klimely, Kurt A. Buhlmann, Brian D. Todd.

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
