## [Decision Letter · Decision Letter 0]

15 Sep 2021

PONE-D-21-24371Habitat use by female desert tortoises suggests tradeoffs between resource use and risk avoidancePLOS ONE

Dear Dr. Nafus,

Thank you for submitting your manuscript to PLOS ONE. After careful consideration, we feel that it has merit but does not fully meet PLOS ONE’s publication criteria as it currently stands. Therefore, we invite you to submit a revised version of the manuscript that addresses the points raised during the review process.

1. You need to better frame the tests of the tradeoff hypothesis in the introduction/methods.2. The discussion should be revised to more explicitly discuss the evidence for and against this hypothesis.3. You need to discuss some alternative explanations for the mortality difference result.4. You need to tune down the language in the discussion, especially in the final few sentences, which currently give the impression that your data strongly support the behavioral tradeoff. ==============================

We look forward to receiving your revised manuscript.

Kind regards,

Ofer Ovadia

Academic Editor

PLOS ONE

Journal Requirements:

“This material is based upon work supported by the National Science Foundation Graduate Research Fellowship Program under Grant No. DGE-1148897 to MGN. A portion of this work was funded by the California Energy Commission’s Public Interest Energy Research (PIER) Program under Agreement # 500-10-020. Manuscript preparation by TDT and KAB was partially supported by the Department of Energy under Award Number DE-FC09- 07SR22506 to the University of Georgia Research Foundation. This work was also supported by the USDA National Institute of Food and Agriculture, Hatch project CA-D-WFB-2097-H.”

We note that you have provided funding information within the Acknowledgements Section. Please note that funding information should not appear in the Acknowledgments section or other areas of your manuscript. We will only publish funding information present in the Funding Statement section of the online submission form.

“National Science Foundation Graduate Research Fellowship Program under Grant No. DGE-1148897 (MGN)

California Energy Commission’s Public Interest Energy Research (PIER) Program under Agreement # 500-10-020 (BTD)

USDA National Institute of Food and Agriculture, Hatch project CA-D-WFB-2097-H (BTD)

Department of Energy under Award Number DE-FC09-07SR22506 (TDT, KAB)

6. Please upload a copy of Supporting Information Table S1 which you refer to in your text on page 11.

Reviewers' comments:

Reviewer's Responses to Questions

**Comments to the Author**

1. Is the manuscript technically sound, and do the data support the conclusions?

Reviewer #1: No

2. Has the statistical analysis been performed appropriately and rigorously? 

Reviewer #1: Yes

3. Have the authors made all data underlying the findings in their manuscript fully available?

Reviewer #1: Yes

4. Is the manuscript presented in an intelligible fashion and written in standard English?

Reviewer #1: Yes

5. Review Comments to the Author

Reviewer #1: In this nice study of G. agassizii comparing two common desert habitat types, Nafus and coauthors provide good evidence of interesting differences in body size, forage availability and burrow site selection. They also quantify survival and above-ground activity as measures of risk-taking behavior. The intro and discussion incorporate appropriate literature and do a good job of placing the study in context. This study is a valuable contribution, but I think the paper could frame the tests of the tradeoff hypothesis more clearly in the introduction/methods, and I have reservations that that they have actually demonstrated differences in risk-taking behavior between the habitat types. The discussion should be revised to more explicitly discuss the presented evidence for (and against) this hypothesis, and discuss alternative explanations for the mortality difference result. The language in the discussion, especially the final few sentences, might make a reader skimming this paper think the evidence supporting a behavioral tradeoff is much more substantial than it is in actuality. By making the ambiguities more transparent in the discussion, and framing the tests more clearly in the intro, the authors can easily make this study acceptable for publication.

Major points:

Stating predictions and explaining tests of hypothesis

The idea that risk taking behavior should trade off with forage availability or nutritional state is clearly explained in the intro, and discussed theoretically at length in the discussion. However, this central pattern is only tangentially addressed in the specific context of the methods and results, and the pertinent results are barely explicitly discussed in the discussion. What exactly do you predict in your results from this theoretical tradeoff? What test was performed to demonstrate this tradeoff? Stating this at the end of the intro would help guide the reader in the methods section.

Interpretation of evidence for risk-taking behavior

As far as I can tell, the available information on risk avoidance behavior is the mortality data, showing that 3 tortoises died in CS while only 1 died in YW, and the surface activity data. Surface activity, an actual measure of tortoise risk-taking behavior, did not differ between the habitat sites. Thus, the conclusion that this study supports the hypothesis that risk taking behavior differs between habitat types rests solely on the barely significant difference in mortality based on rather small sample sizes (though I realize this is a ton of work to assess and difficult to show differences with such long-lived animals!). I think this mixed result should be explicitly spelled out in the discussion, and further caveated with the numerous possible alternative explanations for a mortality difference between habitat types that have nothing to do with behavior (predator abundance differences, tortoise size differences, available forage differences). It's not clear to me how habitat selection alone indicates the tradeoff (L312), so please make this clearer in the Discussion.

L318: I think it's premature to say habitat quality affected morphometric parameters and mortality risk...you merely show an association, not causality.

Given the lack of any behavioral evidence to support it, i think the last sentence is a stretch.

Minor points

L75: could you make explicit predictions from the theoretical tradeoffs described in the intro to guide readers as to how you are addressing your research questions with the data collected? The methods describe data collection but don't describe how the tests performed will address the questions about habitat use framed in the intro.

L102: insert "are" after "Although they..."

L193: specify that this is +/- SE (right?)

L216: change this to "...burrow during 53 ± 4% (range: 20 – 100%) of observations"

L279-80: remove partial sentence here.

L251-256: In G. polyphemus, female-female agression and burrow defense is common. I am not sure whether social behavior is as important in lower-density G. agassizii populations, but is it possible that larger females exclude smaller females from high quality YW habitat?

L251-256: Is there evidence of site/habitat type fidelity on the time scales that would lead to a pattern determined by growth in the juvenile/subadult stage? Or are tortoises moving between habitat types throughout their lives?

6. PLOS authors have the option to publish the peer review history of their article (what does this mean?). If published, this will include your full peer review and any attached files.

Reviewer #1: No

---

## [Author Response · Author response to Decision Letter 0]

8 Dec 2021

1. You need to better frame the tests of the tradeoff hypothesis in the introduction/methods.

We improved the clarity of the tradeoffs being explored using observation data and included our predictions in the final paragraph of the introduction.

2. The discussion should be revised to more explicitly discuss the evidence for and against this

hypothesis.

We revised the discussion to address the reviewer concerns

3. You need to discuss some alternative explanations for the mortality difference result.

We acknowledged that mortality could be caused by exposure or depredation in the discussion. 

4. You need to tune down the language in the discussion, especially in the final few sentences,

which currently give the impression that your data strongly support the behavioral tradeoff. 

We toned down the language related to behavioral tradeoffs.

1. Please ensure that your manuscript meets PLOS ONE's style requirements, including those

for file naming. The PLOS ONE style templates can be found at

https://journals.plos.org/plosone/s/file?

id=wjVg/PLOSOne_formatting_sample_main_body.pdf and

https://journals.plos.org/plosone/s/file?

id=ba62/PLOSOne_formatting_sample_title_authors_affiliations.pdf

We revised the manuscript to follow journal formatting guidelines.

“This material is based upon work supported by the National Science Foundation Graduate

Research Fellowship Program under Grant No. DGE-1148897 to MGN. A portion of this

work was funded by the California Energy Commission’s Public Interest Energy Research

(PIER) Program under Agreement # 500-10-020. Manuscript preparation by TDT and KAB

was partially supported by the Department of Energy under Award Number DE-FC09-

07SR22506 to the University of Georgia Research Foundation. This work was also supported

by the USDA National Institute of Food and Agriculture, Hatch project CA-D-WFB-2097-H.”

We note that you have provided funding information within the Acknowledgements Section.

Please note that funding information should not appear in the Acknowledgments section or

other areas of your manuscript. We will only publish funding information present in the

Funding Statement section of the online submission form.

Please remove any funding-related text from the manuscript and let us know how you would

like to update your Funding Statement. Currently, your Funding Statement reads as follows:

“National Science Foundation Graduate Research Fellowship Program under Grant No. DGE-

1148897 (MGN)

California Energy Commission’s Public Interest Energy Research (PIER) Program under

Agreement # 500-10-020 (BTD)

USDA National Institute of Food and Agriculture, Hatch project CA-D-WFB-2097-H (BTD)

Department of Energy under Award Number DE-FC09-07SR22506 (TDT, KAB)

The funders had no role in study design, data collection and analysis, decision to publish, or

preparation of the manuscript.”

Please include your amended statements within your cover letter; we will change the online

submission form on your behalf.

We deleted the funding information from the acknowledgements.

3. We note that you have stated that you will provide repository information for your data at

acceptance. Should your manuscript be accepted for publication, we will hold it until you

provide the relevant accession numbers or DOIs necessary to access your data. If you wish to

make changes to your Data Availability statement, please describe these changes in your cover

letter and we will update your Data Availability statement to reflect the information you

provide.

We added the information on how to access the data. 

4. Please note that in order to use the direct billing option the corresponding author must be

affiliated with the chosen institute. Please either amend your manuscript to change the

affiliation or corresponding author, or email us at plosone@plos.org with a request to remove

this option.

The corresponding author was affiliated with the institution. 

5. Your ethics statement should only appear in the Methods section of your manuscript. If

your ethics statement is written in any section besides the Methods, please move it to the

Methods section and delete it from any other section. Please ensure that your ethics statement

is included in your manuscript, as the ethics statement entered into the online submission form

will not be published alongside your manuscript.

We moved the ethics statement to the methods section.

6. Please upload a copy of Supporting Information Table S1 which you refer to in your text on

page 11.

A copy of the supporting information was uploaded.

Response to Reviewers

Reviewer #1: In this nice study of G. agassizii comparing two common desert habitat types, Nafus and coauthors provide good evidence of interesting differences in body size, forage availability and burrow site selection. They also quantify survival and above-ground activity as measures of risk-taking behavior. The intro and discussion incorporate appropriate literature and do a good job of placing the study in context. This study is a valuable contribution, but I think the paper could frame the tests of the tradeoff hypothesis more clearly in the introduction/methods, and I have reservations that that they have actually demonstrated differences in risk-taking behavior between the habitat types. The discussion should be revised to more explicitly discuss the presented evidence for (and against) this hypothesis, and discuss alternative explanations for the mortality difference result. The language in the discussion, especially the final few sentences, might make a reader skimming this paper think the evidence supporting a behavioral tradeoff is much more substantial than it is in actuality. By making the ambiguities more transparent in the discussion, and framing the tests more clearly in the intro, the authors can easily make this study acceptable for publication.

Academic Editor

1. You need to better frame the tests of the tradeoff hypothesis in the introduction/methods.

Lines 73-77: We added the following text to better state our intention in the study.

“We sought to determine whether observational patterns in habitat selection by desert tortoises at our study site were consistent with predictions framed under risk tradeoff theories. We predicted that tortoises in high quality forage patches would select habitat that prioritized large shelter plants, whereas tortoises in low quality forage patches would select habitat that prioritized access to nutritious forage plants. The anticipation of a tradeoff between these choices led us to predict increased mortality risk, smaller body size, (a measure of long-term resource availability) and less activity in lower quality habitat (30), while maintaining similar body conditions.”

2. The discussion should be revised to more explicitly discuss the evidence for and against this hypothesis.

Lines 290-304: We have added the following text.

“Female tortoises in this observational study used habitat in a manner consistent with context-dependent risk appraisal given differences in local forage quality. Specifically, tortoises in habitat with less abundant forage used burrows closer to nutritious forage plants than expected at random, whereas tortoises in habitat with abundant forage used burrows more sheltered by large perennials than expected at random. Females in both locations had equivalent body condition scores during the period of study, but not differences in surface activity. The summer foraging season habitat use aligned with subsequent predictions about body size and mortality over the three year period of study, if habitat use patterns were temporally consistent. Repeating this study using populations from other parts of the species’ range would be useful to understand how broadly these results may apply to risk appraisal and habitat use in desert tortoises. Despite being limited to one population within one season of study, the findings here do suggest that risk-taking behavior in wildlife can depend on resource conditions experienced by individuals. By varying the risk associated with habitat selection, female tortoises in low forage habitat were able to maintain similar body conditions as individuals in high forage habitat during the period of study.” 

3. You need to discuss some alternative explanations for the mortality difference result (as well as the different sizes).

Lines 364-365: The following sentence was added.

“Mortality may have been caused by predation or exposure, both of which would be anticipated to increase with tortoises making use of more unsheltered habitat..”

Lines 331-334: We have added the following:

“A An alternative explanation for the observed differences in body size between the habitats could be that females were in greater abundance in one habitat, and territoriality or intra-specific competition may have resulted in slower growth in the low forage habitat. Larger females might exclude smaller females from high quality YW habitat, if territoriality limits co-occupancy.” 

4. You need to tune down the language in the discussion, especially in the final few sentences, which currently give the impression that your data strongly support the behavioral tradeoff. 

Lines 367-383: The last paragraph has been re-written, emphasizing that our limited data, especially on mortality, only suggests risk taking. 

“The observed use of habitat by female tortoises in this study was consistent with expected tradeoffs between resource acquisition and risk avoidance given the differences in forage quality between the two habitats. Our interpretation of the data is that selecting for annual forage—as opposed to prioritizing greater perennial cover—was partly successful for females in habitat with lower forage availability because they were able to maintain similar body conditions to females that had greater access to forage. However, the female tortoises in the lower quality forage habitat were of smaller body sizes and experienced higher mortality than females from the higher forage quality habitat. Although consistent with such a tradeoff, our findings are based on a relatively small number of individuals at a single site over a limited period and are thus limited in scope. In addition, mortality was low during the study period. Thus, further study is needed to support any assertion that habitat quality could influence habitat selection by desert tortoises at fine spatial scales and, in turn, affect morphometric parameters and mortality risk. We argue that female behavior supported that behavioral plasticity can enhance ability to withstand short periods of drought or other resource limitations by prioritizing different habitat characteristics (food or shelter). This study, incipient in nature, should ideally lead others to examine the cost-benefit balance between refuge seeking and foraging activities in a species whose habitat can vary considerably across space and time.”

Major points:

Stating predictions and explaining tests of hypothesis

The idea that risk taking behavior should trade off with forage availability or nutritional state is clearly explained in the intro, and discussed theoretically at length in the discussion. However, this central pattern is only tangentially addressed in the specific context of the methods and results, and the pertinent results are barely explicitly discussed in the discussion. What exactly do you predict in your results from this theoretical tradeoff? What test was performed to demonstrate this tradeoff? Stating this at the end of the intro would help guide the reader in the methods section.

Lines 77-82: We added the following text to better state our intention in the study in the Introduction.

“We predicted that tortoises in high quality forage patches would select habitat that prioritized large shelter plants, whereas tortoises in low quality forage patches would select habitat that prioritized access to nutritious forage plants. The anticipation of a tradeoff between these choices led us to predict increased mortality risk, smaller body size, (a measure of long-term resource availability) and less activity in lower quality habitat (30), while maintaining similar body conditions.

Interpretation of evidence for risk-taking behavior

As far as I can tell, the available information on risk avoidance behavior is the mortality data, showing that 3 tortoises died in CS while only 1 died in YW, and the surface activity data. Surface activity, an actual measure of tortoise risk-taking behavior, did not differ between the habitat sites. Thus, the conclusion that this study supports the hypothesis that risk taking behavior differs between habitat types rests solely on the barely significant difference in mortality based on rather small sample sizes (though I realize this is a ton of work to assess and difficult to show differences with such long-lived animals!). I think this mixed result should be explicitly spelled out in the discussion, and further caveated with the numerous possible alternative explanations for a mortality difference between habitat types that have nothing to do with behavior (predator abundance differences, tortoise size differences, available forage differences). It's not clear to me how habitat selection alone indicates the tradeoff (L312), so please make this clearer in the Discussion.

We have added the following text.

“An alternative hypothesis could be that female were in greater abundance in one habitat, and territoriality or intra-specific competition may have resulted in slower growth in the low forage habitat. A future comparison of growth rates in juvenile tortoises occupying each habitat and the relative availability of food could provide greater insight into a potential mechanism behind the size differences we observed.” 

 and

”However, with the resolution of data we were able to collect, we saw no difference in surface activity by females. In prior work on juveniles, individuals with greater access to water and food spent more time on the surface (30). Thus surface activity may not be a direct measure of risk if not paired with other measures, such as ground temperature or immediate presence of a predator.”

L318: I think it's premature to say habitat quality affected morphometric parameters and mortality risk...you merely show an association, not causality.

Given the lack of any behavioral evidence to support it, i think the last sentence is a stretch.

The last paragraph has been re-written, emphasizing that our limited data, especially on mortality, only suggests risk taking. 

Minor points

L75: could you make explicit predictions from the theoretical tradeoffs described in the intro to guide readers as to how you are addressing your research questions with the data collected? The methods describe data collection but don't describe how the tests performed will address the questions about habitat use framed in the intro.

As addressed in prior comments we made our predictions more explicit. We also directly integrated our findings with our predictions through several sections in the discussion. 

L102: insert "are" after "Although they..."

We have inserted “are” after “Although they…”

L193: specify that this is +/- SE (right?)

This was specified in the methods

L216: change this to "...burrow during 53 ± 4% (range: 20 – 100%) of observations"

We have added “of the observations”.

L279-80: remove partial sentence here.

This paragraph was re-written.

L251-256: In G. polyphemus, female-female agression and burrow defense is common. I am not sure whether social behavior is as important in lower-density G. agassizii populations, but is it possible that larger females exclude smaller females from high quality YW habitat?

This is a really good point. 

Lines 331-341: We have added the following:

“An alternative explanation for the observed differences in body size between the habitats could be that females were in greater abundance in one habitat, and territoriality or intra-specific competition may have resulted in slower growth in the low forage habitat. Larger females might exclude smaller females from high quality YW habitat, if territoriality limits co-occupancy. Female-female aggression and burrow defense have been reported in the closely related gopher tortoise (Gopherus polyphemus) (54). Space use by gopher tortoises—including home range size, overlap, and burrow sharing—also varies as a function of population density (55). It is unknown whether social behavior is as important in G. agassizii populations, which tend to occur at lower densities than gopher tortoises, but it is possible that aggressive behavior may have occurred between females of different sizes. However, in multiple cases during this study the authors also observed females sharing burrows without aggression.” 

L251-256: Is there evidence of site/habitat type fidelity on the time scales that would lead to a pattern determined by growth in the juvenile/subadult stage? Or are tortoises moving between habitat types throughout their lives?

The authors do not know about the site fidelity of tortoises throughout their lives as this has been poorly documented. We do know the both adults and juveniles demonstrate high interannual site fidelity and consistently reuse the same burrows. It’s possible that at some period there is a dispersal age in which they re-position themselves on the landscape, but within the period of this study tortoises were not moving between habitat types. We acknowledged what is known about their site fidelity on LN 310-311.

---

## [Decision Letter · Decision Letter 1]

26 Jan 2022

Habitat use by female desert tortoises suggests tradeoffs between resource use and risk avoidance

PONE-D-21-24371R1

Dear Dr. Nafus,

We’re pleased to inform you that your manuscript has been judged scientifically suitable for publication and will be formally accepted for publication once it meets all outstanding technical requirements.

Kind regards,

Ofer Ovadia

Academic Editor

PLOS ONE

Additional Editor Comments (optional):

Reviewers' comments:

Reviewer's Responses to Questions

**Comments to the Author**

1. If the authors have adequately addressed your comments raised in a previous round of review and you feel that this manuscript is now acceptable for publication, you may indicate that here to bypass the “Comments to the Author” section, enter your conflict of interest statement in the “Confidential to Editor” section, and submit your "Accept" recommendation.

Reviewer #1: All comments have been addressed

2. Is the manuscript technically sound, and do the data support the conclusions?

Reviewer #1: Yes

3. Has the statistical analysis been performed appropriately and rigorously? 

Reviewer #1: Yes

4. Have the authors made all data underlying the findings in their manuscript fully available?

Reviewer #1: Yes

5. Is the manuscript presented in an intelligible fashion and written in standard English?

Reviewer #1: Yes

6. Review Comments to the Author

Reviewer #1: The authors have addressed my concerns and I believe the manuscript is now acceptable for publication.

7. PLOS authors have the option to publish the peer review history of their article (what does this mean?). If published, this will include your full peer review and any attached files.

Reviewer #1: No

---

## [Editor Report · Acceptance letter]

28 Jul 2022

PONE-D-21-24371R1 

Habitat use by female desert tortoises suggests tradeoffs between resource use and risk avoidance 

Dear Dr. Todd:

I'm pleased to inform you that your manuscript has been deemed suitable for publication in PLOS ONE. Congratulations! Your manuscript is now with our production department. 

Kind regards, 

on behalf of

Dr. Ofer Ovadia 

Academic Editor

PLOS ONE